# Prototype Optical Bionic Microphone with a Dual-Channel Mach–Zehnder Interferometric Transducer

**DOI:** 10.3390/s23094416

**Published:** 2023-04-30

**Authors:** Xin Liu, Chen Cai, Kangning Ji, Xinyu Hu, Linsen Xiong, Zhi-mei Qi

**Affiliations:** 1State Key Laboratory of Transducer Technology, Aerospace Information Research Institute, Chinese Academy of Sciences, Beijing 100190, China; 2School of Electronic, Electrical, and Communication Engineering, University of Chinese Academy of Sciences, Beijing 100049, China; 3School of Optoelectronics, University of Chinese Academy of Sciences, Beijing 100049, China

**Keywords:** optical directional microphone, Mach–Zehnder interferometer, bionic MEMS silicon diaphragm, temperature robustness, high sensitivity

## Abstract

A prototype optical bionic microphone with a dual-channel Mach–Zehnder interferometric (MZI) transducer was designed and prepared for the first time using a silicon diaphragm made by microelectromechanical system (MEMS) technology. The MEMS diaphragm mimicked the structure of the fly Ormia Ochracea’s coupling eardrum, consisting of two square wings connected through a neck that is anchored via the two torsional beams to the silicon pedestal. The vibrational displacement of each wing at its distal edge relative to the silicon pedestal is detected with one channel of the dual-channel MZI transducer. The diaphragm at rest is coplanar with the silicon pedestal, resulting in an initial phase difference of zero for each channel of the dual-channel MZI transducer and consequently offering the microphone strong temperature robustness. The two channels of the prototype microphone show good consistency in their responses to incident sound signals; they have the rocking and bending resonance frequencies of 482 Hz and 1911 Hz, and their pressure sensitivities at a lower frequency exhibit an “8”-shaped directional dependence. The comparison indicates that the dual-channel MZI transducer-based bionic microphone proposed in this work is advantageous over the Fabry–Perot interferometric transducer-based counterparts extensively reported.

## 1. Introduction

Sound source localization (SSL) is an important technique with widespread applications in different scenarios such as the positioning of naval gunfire, tracking of unmanned aerial vehicles, detection of pipeline leakage, and auditory navigation for mobile robots [1,2,3,4,5,6,7,8,9,10]. SSL is typically achieved using microphone arrays based on evaluation of the difference in time or intensity between the two microphones at which the sound arrives. To prevent the phase differences from being blurred and to achieve accurate SSL, the distance between adjacent microphones in a microphone array is required to be as long as possible, within half the wavelength of the incident sound [11]. The wavelength of sound waves is large, making it difficult to miniaturize microphone arrays. To meet the portable applications of SSL, it is necessary to develop a miniature device with a distinctive SSL mechanism.

Twenty years ago, scientists discovered in nature a small parasitic fly called Ormia Ochracea that has an excellent SSL ability that is not commensurate with its size [12,13]. The fly Ormia Ochracea has a binaural space of less than 0.5 mm, while it can localize its host cricket through hearing the sound of about 7 cm wavelength from the host [12,13,14,15,16,17,18,19]. Anatomical analysis reveals that the binaural tympanic membranes of the fly Ormia Ochracea are connected by a rigid cuticle, thus forming a mechanically coupled structure. The coupled structure of the Ormia Ochracea fly’s ear can significantly amplify the interaural intensity difference (IID) and interaural time difference (ITD) [13,14]. Therefore, the fly Ormia Ochracea, with very small binaural space, could achieve SSL with an angular range of ±30° and an accuracy of ±2° [19,20,21]. Inspired by Ormia Ochracea’s mechanically coupled ears, bionic microphones can be made small enough to be used as portable SSL devices. Driven by potentially significant demand, such bionic microphones have been extensively studied; their diaphragms are often prepared with MEMS technology through mimicking the structure of the fly Ormia Ochracea’s ear. According to the literature, the vibrational displacement of the diaphragms of these bionic microphones are read out by comb capacitance [22,23,24,25], piezoelectric [26,27,28,29,30,31], and optical [20,21,32,33,34] methods. The electrical bionic microphones have two common drawbacks: (1) they are susceptible to electromagnetic interference; (2) their sensitivity is low due to the small capacitance related to the small diaphragm. These disadvantages limit the application of electrical bionic microphones.

Optical bionic microphones can overcome the above disadvantages of electrical bionic microphones. Most of the optical bionic microphones reported so far are based on the Fabry–Perot interferometer (FPI) transducer. Liu et al. [20,21] and Ren et al. [34] designed and prepared FPI-based fiber-optic bionic microphones with high sensitivity and high SSL accuracy within a narrow range of frequency. However, the performance of FPI-based fiber-optic bionic microphones can be easily affected by environmental temperature due to the presence of the FPI cavity that is formed between the reflective surface of the diaphragm and the fiber end face. The initial length of the FPI cavity varies with temperature, causing the operating point of the microphone to drift. Although a short FPI cavity is conducive to improving the thermal stability of the microphones to a certain extent [35], it is difficult to completely eliminate the thermal drift of its operating point. It is critical to develop thermally stable optical bionic microphones by using an intrinsically temperature-insensitive optical transducer, such as the Mach–Zehnder interferometer (MZI).

In this work, we designed a novel prototype optical microphone based on the combination of a dual-channel MZI transducer and a bionic MEMS silicon diaphragm. MZI is a two-beam interferometer: one beam is used for sensing, and the other beam serves as the reference. Since the two beams are very close to each other and have an equal path length, the MZI has an initial phase difference of zero and is immune to temperature fluctuation. The bionic MEMS diaphragm consists of two square wings connected through a neck that is anchored via the two torsional beams to the silicon base. The designed prototype optical bionic microphone uses a single laser beam to strike the area containing the diaphragm edge and the silicon base edge; the light reflected from the diaphragm edge is the sensing component, and that which is reflected from the base edge is the reference component. With a lens, the reflected beam is focused on the photodetector to yield interference between the sensing and reference components. The diaphragm and base are coplanar, and thus the MZI has an initial phase difference of zero. Therefore, the designed bionic optical microphone would be resistant to temperature influence while keep the acoustic pressure sensitivity equal to that measured with the FPI transducer.

## 2. Methodology

The MEMS diaphragm mimics the structure of the fly Ormia Ochracea’s coupling eardrum and was prepared with a silicon-on-insulator (SOI) substrate. As shown in Figure 1a, the bionic diaphragm consists of two square wings (2 mm × 2 mm), a square coupling bridge (1 mm × 1 mm), and two torsion legs (0.1 mm × 0.2 mm). The two wings are connected by the coupling bridge, and the two torsion legs are used to fix the coupling bridge onto the SOI base. An air gap between the diaphragm and the SOI base is designed to be 100 μm in width. The thickness of the bionic MEMS diaphragm is 10 μm. The vibrational displacement of each wing at its distal edge relative to the SOI base is detected in real-time with one channel of the dual-channel MZI transducer.

The designed prototype optical bionic microphone uses a dual-channel MZI transducer to simultaneously detect the vibrational displacements at the distal edges of both wings of the diaphragm. Due to the symmetry of structure of the bionic MEMS diaphragm, the two MZI channels have an identical optical arrangement. Figure 1b schematically shows the optical arrangement of a single-channel MZI transducer. A laser beam passes through a semi-reflective/semi-transmissive mirror and illuminates the designated area of the bionic MEMS diaphragm at normal incidence. The illuminated area contains the edge of the SOI base, the edge of a wing of the diaphragm, and the air gap between the two edges. The two partial beams of light are reflected back to the mirror by the reflective surface of the SOI base and that of the wing and overlap on a photoreceptor through the reflection of the mirror and the subsequent focusing of the lens. As shown in Figure 1c, the partial beam reflected by the wing is used for sensing the vibration of the diaphragm, and that which is reflected by the SOI base serves as the reference. The interference of the two partial beams occurs on the photodetector, and the detected light intensity can be expressed as (1) [36]:(1)$$I=I_{S}+I_{R}+2\sqrt{I_{S}I_{R}}\cos(\frac{4\pi d}{\lambda })$$
where $I_{R}$ and $I_{S}$ are the intensities of the reference and sensing light beams; λ is the wavelength of the light source used; and *d* is the vibrational displacement of the wing of the diaphragm at the observed point.

## 3. Simulation of Characteristics of the Bionic Diaphragm

### 3.1. Rocking-Mode and Bending-Mode Resonance Frequencies of the Bionic Diaphragm

As shown in Figure 2a, a three-dimensional (3D) model was created with the designed bionic MEMS diaphragm located in the acoustic field emitted from a speaker, for simulation of the vibrational characteristics of the diaphragm based on the finite element method. In Figure 2a, the x and z axes are the long and short axis of the bionic diaphragm, and the y axis is normal to the diaphragm. In the finite element simulation process, the SOI base surrounding the bionic MEMS diaphragm was set to be a fixed boundary. We first simulated the vibration modes of the designed bionic MEMS diaphragm. As shown in Figure 2b, the bionic MEMS diaphragm has two vibration modes, namely rocking mode and bending mode. The rocking mode is the antisymmetric vibration mode, which is sensitive to the sound pressure gradient, and the bending mode is the symmetrical vibration mode and sensitive to the sound pressure. Therefore, only the rocking mode of the bionic diaphragm has directional sensitivity and thus offers the diaphragm the SSL capability at a frequency within the rocking-mode resonance band. These features of the designed bionic diaphragm are exactly as same as the vibrational characteristics of the fly Ormia Ochracea’s ear.

Next, the frequency response of the bionic MEMS diaphragm was simulated. In the simulation model, the incident sound is a planar wave propagating along the x axis (namely, *θ* = 90º in Figure 2a), and the sound pressure is 1 Pa. The effect of air damping on the vibration of the bionic diaphragm was considered in the simulation. The amplitudes at the distal edges of both wings of the bionic MEMS diaphragm were calculated simultaneously, while the frequency of the acoustic wave varied from 200 Hz to 2 kHz with a frequency interval of 20 Hz. Since the frequency responses of the two wings are identical to each other, the simulation result with one wing is shown in Figure 2c, where the two peak frequencies are 820 Hz and 1590 Hz, corresponding to the resonance frequencies of the rocking and bending modes of the diaphragm, respectively.

### 3.2. Directional Dependence of the Mechanical Sensitivity of the Bionic Diaphragm

As shown in Figure 2a, a planar sound wave propagates parallel to the *x*–*y* plane and is incident on the bionic diaphragm at an angle *θ*. The *x*-axis component of the planar sound wave can be represented as Equation (2) [27]:(2)$$p(t,x)=p_{0}e^{j\omega \left[t+\frac{1}{c}x\sin\left(\theta \right)\right]}$$
where $p_{0}$ is the amplitude of the sound wave; *j* is an imaginary unit; *ω* is the angular frequency of the sound; *t* is time; and *c* is the speed of sound in air. By processing Equation (2) with the first-order Taylor series expansion, the *x*-axis component of the planar sound wave can be approximated as Equation (3):(3)$$p\left(t,x\right)={\left.p\left(t,x\right)\right|}_{x=0}+x{\left.\frac{\partial p\left(t,x\right)}{\partial x}\right|}_{x=0}=p_{0}e^{j\omega t}+xp_{0}e^{j\omega t}\frac{j\omega }{c}\sin(\theta )$$

In rocking mode with the rocking-mode resonance band, the displacement of the bionic MEMS diaphragm is dependent only on the pressure gradient of the x-component of the incident wave [27]. The displacement of the bionic MEMS diaphragm can be expressed as:(4)$$d\propto \left|\frac{\partial p(t,x)}{\partial x}\right|=\left|p_{0}e^{j\omega t}\frac{j\omega }{c}\sin(\theta )\right|\propto \left|\sin(\theta )\right|$$

Equation (4) reveals that the displacement at any point of the bionic MEMS diaphragm is proportional to the absolute value of the sin(*θ*), which presents an “8” shape in the polar coordinate system. This relationship between the displacement of the bionic MEMS diaphragm and the incident angle of the sound wave indicates that the SSL within a certain angular range could be achieved using the bionic diaphragm.

We further calculated the directional response of the bionic MEMS diaphragm to an incident sound wave with a pressure of 1 Pa and a frequency equal to the rocking-mode eigenfrequency (820 Hz). Figure 2d shows the simulation results in the polar coordinate system. A plot of the displacement at the distal end of the diaphragm against the incident angle of the sound shows a standard “8”-shaped response. The displacement monotonically varies with the incident angle of the sound wave in a range from 0 to 90°, verifying the SSL capability within a certain angular range for the designed bionic diaphragm.

## 4. Experimental

### 4.1. Fabrication Process of the Bionic MEMS Diaphragm

The substrate used is a 4-inch SOI wafer containing a 10 µm thick device layer, a 1 µm thick buried oxide layer, and a 360 µm thick handle layer. The bionic MEMS diaphragm is fabricated according the following steps: (a) A photoresist layer is deposited on the device layer and then patterned through a mask with the bionic diaphragm pattern (Figure 3a). (b) The exposed device layer is etched with deep reactive ion etching (DRIE) (Figure 3b). (c) The etched device layer is protected with a layer of viscous photoresist to prevent its being broken by the stresses released during the subsequent DRIE (Figure 3c). (d) The handle layer is etched with DRIE after patterning the photoresist layer coated on it, and the etching stops at the buried oxide layer (Figure 3d). (e) The buried oxygen layer is removed from the backside of the wafer using reaction ion etching (RIE) (Figure 3e). (f) A 20 nm thick chromium layer is sputtered on the etched device layer to increase the reflectivity of the bionic MEMS diaphragm for improving sensitivity of the optical bionic microphone (Figure 3f).

The prepared bionic MEMS diaphragm was characterized with a scanning electron microscope (SEM). Figure 4 displays the SEM image of the diaphragm. The sizes of microstructures of the bionic MEMS diaphragm are identical to the designed sizes. The SEM image shows that the diaphragm is smooth, and the wings do not bend upwards or downwards. The findings indicate that the wings composed of the diaphragm are coplanar with the surrounding SOI base.

### 4.2. Construction of the Prototype Optical Bionic Microphone

The principle prototype of the dual-channel MZI-based optical bionic MEMS microphone was constructed on a small optical baseplate using a 633 nm helium–neon laser (SPL Photonics Co, SPL-HN, Hangzhou, China), two beam splitters (BS) (Thorlabs, PBS103, Newton, NJ, USA), a glass prism (Thorlabs, MRA10L-E02, Newton, NJ, USA), two small lenses, and two photodetectors mounted on a PCB. In addition, some optical holders and stages were also used. For ease of understanding, the designed optical arrangement of the prototype microphone is shown in Figure 5a. The first BS (BS1) and the prism combine into the BS–prism module that not only splits the laser beam into two parallel beams, but also enables control of the inter-beam distance to match the spacing between the distal edges of the two wings of the bionic MEMS diaphragm. The two parallel beams are used to realize a dual-channel MZI transducer to simultaneously detect displacements of both wings of the bionic MEMS diaphragm at their distal edges. The laser beam has a diameter of ca. 1 mm, ensuring that the illuminated area of the diaphragm contains the edge of the SOI base, the edge of a wing, and the air gap between the two edges. Figure 5b shows a photograph of the prototype dual-channel MZI-based optical bionic MEMS microphone prepared for principle validation. It can be seen from the photograph that there are two optical baseplates connected through a rotating stage, and the top baseplate overlaid with the prototype microphone can rotate relative to the bottom baseplate. This rotating structure is used for investigating the directional response of the prototype microphone.

### 4.3. Test Platform of the Prototype Optical Bionic Microphone

To test the frequency response of the prototype optical bionic microphone, a reference microphone (B&K, 4193-L-004, DK-2830 Virum, Denmark) was mounted on the top baseplate to be close to the bionic MEMS diaphragm (see Figure 5b). A speaker (Visaton, FRS 5X-8 Ohm, Kowloon, Hong Kong, China) was connected with a generator module (B&K, 3160-A-042, DK-2830 Virum, Denmark) and was positioned in front of the reference microphone and the bionic diaphragm. The distances from the loudspeaker to the reference microphone were equal to that of the distance to the bionic diaphragm. The sweep sound waves from the loudspeaker were incident on the reference microphone and the bionic diaphragm at θ ≈ 0°. The signal output from the reference microphone and the two MZI channels of the prototype bionic microphone were analyzed with the B&K PULSE Labshop software.

To compare the behavior of temporal response to a single-frequency sound between the two MZI channels of the prototype optical bionic microphone, the experimental measurements were performed in a conference room. The sound source used was a home speaker (TOneWinner, TDB-09, Guangzhou, China) with a power amplifier (TOneWinner, AD-300K, Guangzhou, China) that was connected with a signal generator (Agilent, 33500B, Beijing, China). The output signals from both MZI channels of the prototype microphone are acquired through a data acquisition card (National Instruments, NI-USB-4431, Austin, TX, USA) and the signals are then processed with the LabVIEW software (National Instruments, LabVIEW 2016, Austin, TX, USA). Using the same test system, the directional response of the prototype optical bionic microphone was investigated in an anechoic chamber. Figure 6 shows the experimental scenario. Note that in all measurements, the distance between the home speaker and the bionic MEMS diaphragm is about 1.2 m.

## 5. Results and Discussion

### 5.1. Frequency Response of the Prototype Optical Bionic Microphone

Figure 7 shows the two frequency response curves obtained with the two MZI channels of the prototype optical bionic microphone (@ θ = 0°). Each curve contains two resonant peaks located at 482 Hz and 1911 Hz, corresponding to the rocking-mode and bending-mode resonance frequencies. The experimental results indicate that the frequency responses of the two MZI channels are consistent. However, the measured frequencies are quite different from the simulation data (820 Hz for the rocking mode and 1590 Hz for the bending mode), mainly attributable to three reasons. The first reason is sputtering of the chromium thin film on the bionic MEMS diaphragm. This layer was not considered during the simulation process. The second reason is that the MEMS processing induced residual stress in the diaphragm, and the third reason is small deviation in the size of the actual chip from the design size.

### 5.2. Comparison of the Response Behavior of the Two MZI Channels of the Prototype Optical Bionic Microphone

Figure 8a shows the temporal response of the two MZI channels of the prototype microphone to a lower-frequency sound wave incident at θ = 0°. The sound pressure is 0.5 Pa, and its frequency is 482 Hz, equal to the rocking-mode resonance frequency. Evidently, the output signals of both MZI channels are opposite in phase. Figure 8a also shows the response amplitudes in frequency domain for the two MZI channels, which were obtained by processing the measured temporal response data through fast Fourier transform (FFT) method. The response amplitudes of the two MZI channels are very close to each other. In addition, the temporal responses to a higher-frequency sound and the corresponding response amplitudes in frequency domain for the two MZI channels of the prototype microphone were also investigated. The sound pressure is 1.23 Pa, and its frequency is 1911 Hz, equal to the bending-mode resonance frequency. Figure 8b displays the measured results. In this case, the output signals of both MZI channels of the prototype microphone have exactly the same phase and almost equal amplitude. The findings demonstrated that the MZI transducer can indeed significantly improve the response consistency of both channels of the prototype bionic microphone.

As shown in Figure 8a,b, the response amplitudes obtained by FFT method is precisely located at the frequencies of the incident sound waves. From the FFT results, the signal–noise ratio (SNR) of both MZI channels of the prototype microphone can be obtained. At the rocking-mode frequency, the SNR is 64.9 dB for CH1 and 63.9 dB for CH2. At the bending-mode frequency, the SNR is 55.6 dB for CH1 and 55.9 dB for CH2. By submitting the above SNR values into the following equation [34], the minimum detectable pressures (MDP) for CH1 and CH2 of the prototype microphone were calculated.
(5)MDP=P10SNR20×∆f
where ∆f is the frequency resolution given in the FFT process and ∆f is 2 Hz in this work, and *P* is the sound pressure. The calculated results show that at the rocking-mode frequency, the MDP is 0.142 mPa/√Hz for CH1 and 0.16 mPa/√Hz for CH2 and at the bending-mode frequency, the MDP is 1.02 mPa/√Hz for CH1 and 0.99 mPa/√Hz for CH2. Although the MDP value obtained at the rocking-mode frequency is smaller than the previously reported MDP value for the FPI-based fiber optic bionic microphone (1.5 mPa/√Hz at its rocking mode frequency and θ = 0°) [34], this does not mean that this prototype microphone is capable of detecting faint sounds, because its MDP is still large relative to the existing electret condenser microphones. The relative large MDP values for the present prototype microphone mainly results from the discrete bulky optical elements used. It would be possible to significantly improve the MDP of the prototype microphone by designing the dual-channel MZI transducer into a compact module.

### 5.3. Directional Response of the Prototype Optical Bionic Microphone

The directional responses for the two MZI channels of the prototype optical bionic microphone were measured by step-by-step changing the incident angle of the sound under the condition that the power of the incident sound is given to be 58 dB (relative to 20 μPa) and its frequency is fixed at the rocking-mode resonance frequency of the diaphragm (482 Hz). Figure 9a shows the measured response amplitudes versus the incident angle in the polar coordinate system, which results in a “8”-shape pattern for each MZI channel. The measured directional response of the prototype microphone is quite similar to the simulation result shown in Figure 2d. The two “8” patterns in Figure 9a do not overlap exactly, which is not due to measurement error, but implying a slight difference in directional response between the two MZI channels of the prototype microphone. This slight difference between the two wings of the bionic diaphragm was also observed in the simulation results (not shown). Plotting in the Cartesian coordinate system the response amplitudes measured with each MZI channel against the incident angles of the sound wave leads to a sinusoidal curve, as shown in Figure 9b. It is known from Figure 9b that the linear dependence of the microphone response amplitude on the incident angle exists in the narrow angle range within 60º. Based on this linear relationship, the prototype bionic microphone is capable of achieving precise SSL over an angle range of 60°.

## 6. Conclusions

This work demonstrates the first successful application of the MZI transducer for realization of an optical bionic microphone. The MZI transducer is highly compatible with the MEMS bionic diaphragm, and it is designed to easily detect the displacement at the distal edge of the diaphragm, which is larger than those at other positions of the diaphragm. The MZI can convert the displacement at the diaphragm inspection point into half the optical path difference between the sensing and reference beams. As a result, the MZI-based optical bionic microphone has high sensitivity to sound signal. When the diaphragm is at rest, the phase difference between the sensing and reference beams of the MZI is zero, making the bionic microphone robust to temperature fluctuation. Using the discrete optical elements and a laboratory-made SOI-based MEMS bionic diaphragm, a prototype optical bionic microphone with a dual-channel MZI transducer was prepared to simultaneously detect the two displacements at both opposite distal edges of the diaphragm. The experimental results show that the output signals from the two MZI channels is opposite in phase and equal in amplitude when the frequency of sound wave is within the rocking-mode resonance band of the bionic diaphragm and that the response amplitude of each MZI channel varies with the incident angle of the sound wave, forming an “8” pattern in the polar coordinate system. This work shows the significant advantages and high applicability of MZI transducer technology in developing miniaturized high-performance optical bionic microphones for SSL applications.

## Figures and Tables

**Figure 1 sensors-23-04416-f001:**
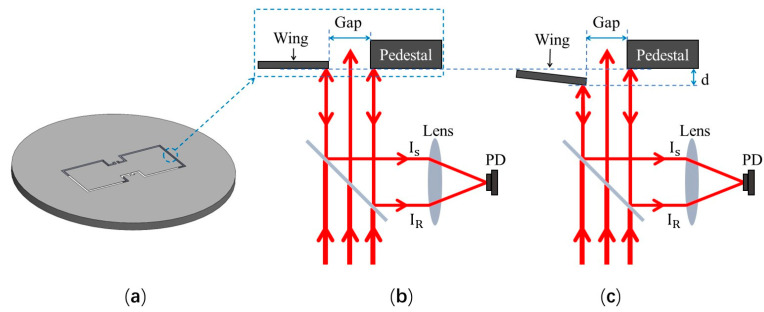
Schematic illustration of detection principle of the MZI-based prototype optical bionic microphone: (**a**) the bionic MEMS diaphragm designed in this work; (**b**,**c**) MZI designed for detecting the displacement at the distal edge of the bionic diaphragm ((**b**,**c**): without and with sound excitation).

**Figure 2 sensors-23-04416-f002:**
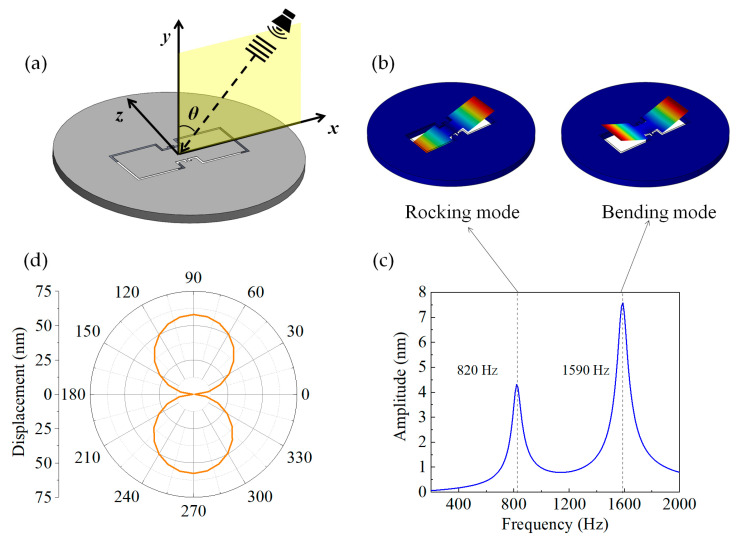
Simulation results for the bionic MEMS silicon diaphragm: (**a**) simulation model; (**b**) rocking and bending modes of the diaphragm; (**c**) frequency response curve; (**d**) the “8”-shaped directional dependence of the displacement at the distal edge of the diaphragm obtained at the sound frequency of 820 Hz and the sound pressure of 1 Pa.

**Figure 3 sensors-23-04416-f003:**
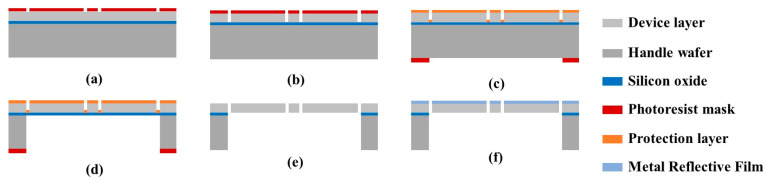
MEMS process for fabricating the bionic diaphragm on SOI substrate.

**Figure 4 sensors-23-04416-f004:**
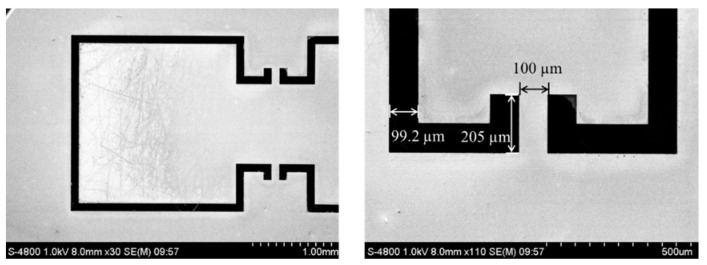
SEM images of the bionic MEMS diaphragm fabricated in laboratory.

**Figure 5 sensors-23-04416-f005:**
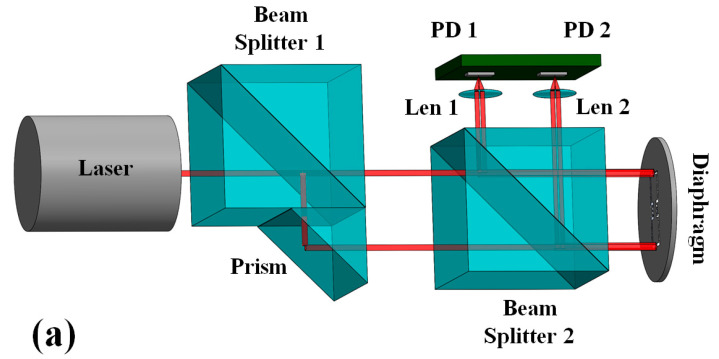
The dual-channel MZI based prototype optical bionic microphone, (**a**) Schematic diagram of the prototype optical bionic microphone; (**b**) Photograph of the actual prototype microphone (the experimental system for frequency response measurement is also schematically shown).

**Figure 6 sensors-23-04416-f006:**
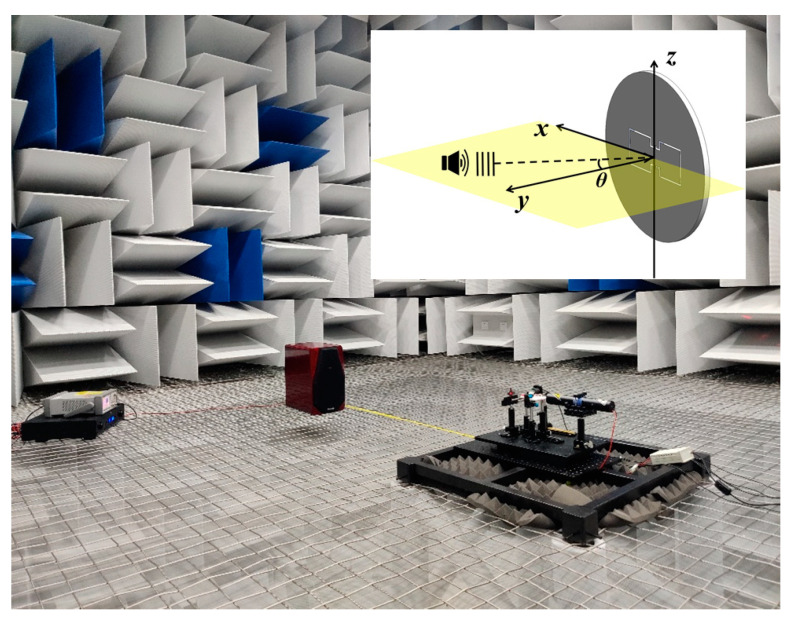
Experimental setup for measuring the directional response of the MZI-based prototype optical bionic microphone in the anechoic room.

**Figure 7 sensors-23-04416-f007:**
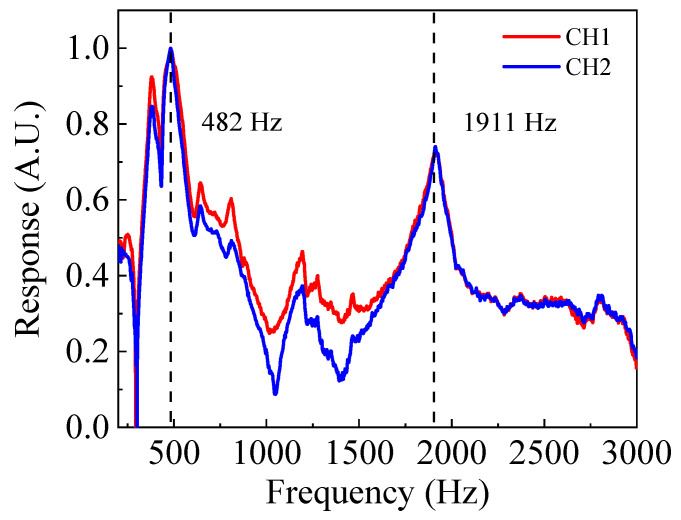
Normalized frequency response for both MZI channels (CH1 and CH2) of the prototype optical bionic microphone.

**Figure 8 sensors-23-04416-f008:**
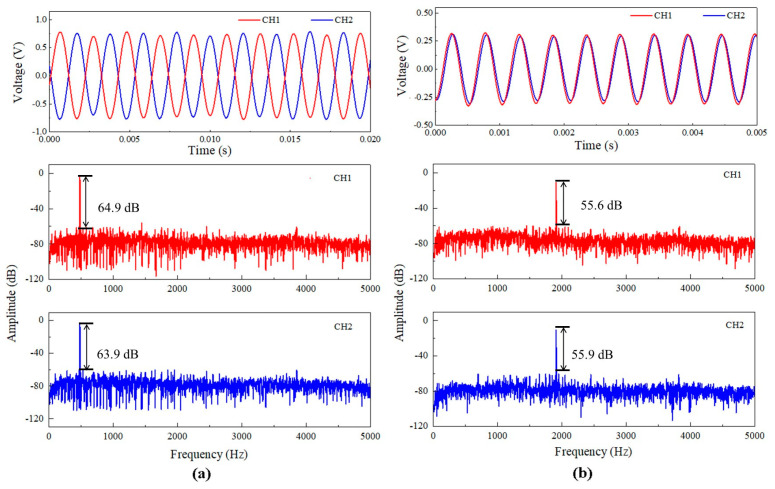
Measured temporal responses for both MZI channels of the prototype optical bionic microphone and the corresponding response amplitudes in frequency domain obtained by FFT processing of the temporal response data: (**a**) the incident sound at the rocking-mode resonance frequency (482 Hz); (**b**) at the bending-mode resonance frequency (1911 Hz).

**Figure 9 sensors-23-04416-f009:**
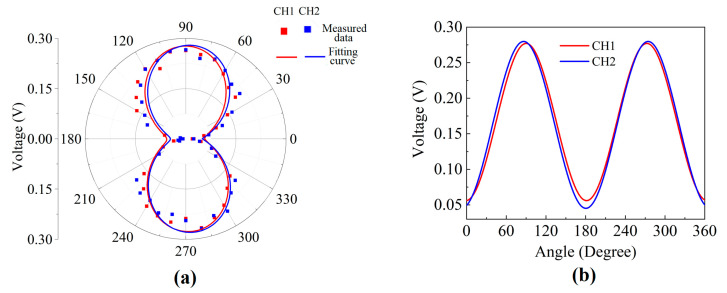
Directional responses to a sound wave at the rocking-mode frequency measured with both MZI channels (CH1 and CH2) of the prototype microphone: (**a**) the “8” patterns in the polar coordinate system; (**b**) the sinusoidal curves in the Cartesian coordinate system.

## Data Availability

The data obtained in this work are available from the corresponding author upon reasonable request.

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
