# Peer review of "Prototype Optical Bionic Microphone with a Dual-Channel Mach–Zehnder Interferometric Transducer"

_sensors, 2023, doi:10.3390/s23094416_

Round 1

Reviewer 1 Report

the inner band flatness of the microphone is not good, the band width is narrow, on the technical level achieved in this paper, the practicality is limited.

Reviewer 2 Report

This paper reports a directional microphone composed of a MEMS beam inspired by the fly ormia ochracea, and the Mach-Zehnder interferometer sensing system. The directionality is calculated analytically, simulated with FEM, and proved experimentally. Optical microphones based on Fabry-Perot interferometer are popular while there are not many reports of Mach-Zehnder interferometer based microphones.  Therefore, it is good to see this paper present the directional microphone with Mach-Zehnder interferometer.

However, there are some flaws in the manuscript making the contribution and important information not clear, which should be revised.

For overall recommendation, if the manuscript is revised properly, it will be suited to be published. My detailed questions/ comments/suggestions on the content are in the attached PDF file. Thank you.

Reviewer 3 Report

The paper propose an innovative MEMS microphone based on a Dual-channel 2 Mach-Zehnder Interferometric Transducer. Authors gives a very clear description of the system and of the results. I have only some minor suggestions to improve the overall work.

First, the state of the art is not so complete. In particular, some key aspects such as the IID and the ITD should be better explained and a reference should be added. In general, the number of reference is very limited for a journal paper.

Regarding the simulated scenario some additional information should be given, e.g., the software used for the finite element simulation and some parameters used for setting up the simulation.

On line 246 authors speaks about a generic NiDaq-MX acquisition board, but a more detailed description of the acquisition system should be given. What Daq has been used, which Ni Chassis has been used, which version of Labview ?

The conclusion doesn’t give any information regarding the future works. I think that some consideration on possible real word scenarios for this microphone could be added. Also some considerations on costs and miniaturization of the system could be interesting.

Round 2

Reviewer 2 Report

The authors addressed the questions, comments, and suggestions raised by the reviewers. The manuscript is good to be published.